# Toxic Effects of TiO_2_ NPs on Zebrafish

**DOI:** 10.3390/ijerph16040523

**Published:** 2019-02-13

**Authors:** Tianle Tang, Zhang Zhang, Xiaopeng Zhu

**Affiliations:** 1School of Tropical Medicine and Laboratory Medicine, Hainan Medical University, Haikou 570228, China; tianle_tang@hainmc.edu.cn; 2Key Laboratory of Tropical Biological Resources, Ministry of Education, Hainan University, Haikou 570228, China; zzhang112233@126.com

**Keywords:** TiO_2_ NPs, zebrafish, oxidative damage, quantitative real-time reverse transcription PCR (qRT-PCR)

## Abstract

Titanium dioxide nanoparticles (TiO_2_ NPs) have become a widely used nanomaterial due to the photocatalytic activity and absorption of ultraviolet light of specific wavelengths. This study investigated the toxic effects of rutile TiO_2_ NPs on zebrafish by examining its embryos and adults. In the embryo acute toxicity test, exposure to 100 mg/L TiO_2_ NPs didn’t affect the hatching rate of zebrafish embryos, and there was no sign of deformity. In the adult toxicity test, the effects of TiO_2_ NPs on oxidative damage in liver, intestine and gill tissue were studied. Enzyme linked immunosorbent assay (ELISA) and fluorescence-based quantitative real-time reverse transcription PCR (qRT-PCR) were used to detect the three antioxidant enzymes: superoxide dismutase (SOD), catalase (CAT) and glutathione S transferase (GSTs) in the above mentioned zebrafish organs at protein and gene levels. The results showed that long-term exposure to TiO_2_ NPs can cause oxidative damage to organisms; and compared with the control group, the activity of the three kinds of enzyme declined somewhat at the protein level. In addition, long-term exposure to TiO_2_ NPs could cause high expression of CAT, SOD and GSTs in three organs of adult zebrafish in order to counter the adverse reaction. The effects of long-term exposure to TiO_2_ NPs to adult zebrafish were more obvious in the liver and gill.

## 1. Introduction

TiO_2_ can form a nanocrystalline structure with a high specific surface area, i.e., titanium dioxide nanoparticles (TiO_2_ NPs). TiO_2_ NPs mainly exist in two forms-Rutile and Anatase. With the photocatalytic activity and ultraviolet absorption activity at specific wavelengths, TiO_2_ NPs have become a widely used nanomaterial, especially in the cosmetics industry [1]. Compared with ordinary TiO_2_, TiO_2_ NPs present better transparency on human skin and are therefore largely used in cosmetic products like sunblock etc. As released by United States Environment Agency, the annual yield of TiO_2_ NPs reaches 2000 ton in 2009 in the United States, among which 65% has been used in cosmetics and sunblock [2]. Besides, as TiO_2_ NPs have antibacterial and organic catalytic functions, they can also be used for sterilization and the degradation of organisms, as well as in other industries like inks and self-cleaning ceramics, glass, paint, and papermaking [3].The yearly demand of nano-TiO_2_ particles is estimated to hit 2.5 × 10^6^ tons by 2025 in the United States [4]. Currently, no restrictions have been set on cosmetics. In China, the suggested number for cosmetics is about 5%. The increasing manufacture and incorporation of nano-TiO_2_ in various applications will inevitably lead to release into the environment, thus causing pollution and imposing a potential hazard to ecological safety.

As TiO_2_ NPs leakage through air, soil and water is unavoidable during production, usage and waste disposal, the ecological effects to the natural environment have aroused extensive attention both domestically and internationally. In recent years, various research has gradually unveiled the fact that TiO_2_ NPs are not as non-toxic and harmless as people used to believe. TiO_2_ NPs can trigger oxidative damage of free radicals and oxidative stress response, cell membrane damage, mitochondria and other organelle damage, key protein inactivation, chromosome damage and DNA damage etc. Wu found that certain amount of TiO_2_ NPs could penetrate the corneum of pigs and reach the deeper layer of the epidermis [5]. In 2018, researchers found that the exposure of certain amount of TiO_2_ NPs to skin could cause damage to the brain of a mouse [6]. In some other research about plants, triggered oxidative stress response in rice through the absorption of TiO_2_ NPs via the root has also been discovered. TiO_2_ NPs could affect the formation of iron film in the root system of rice and hence hinder the absorption of minerals like Fe, Mn and Zn, suppressing the growth of rice [7].

Therefore, careful research and evaluation on biological safety of TiO_2_ NPs are necessary. As a model organism, zebrafish is extensively used in the research of neurodevelopmental biology as well as the evaluation of drug safety and ecotoxicology [8,9]. Through acute and long-term TiO_2_ NPs exposure experiments on the embryo and adult zebrafish, this research examines the ecotoxicological effects of TiO_2_ NPs on zebrafish, systematically investigating the toxic effects of TiO_2_ NPs on different tissues and organs of zebrafish at embryonic, cellular and molecular levels.

## 2. Materials and Methods

### 2.1. Reagents, Equipment and Laboratory Animal

TiO_2_ (≥99% purity) and Rutile-TiO_2_ (model T104942) (with an average diameter of 25 nm; ≥99.8% purity) were purchased from Shanghai Aladdin biochemical Polytron Technologies Inc (Shanghai, China); Superoxide Dismutase (SOD), Glutathione S-transferase (GSTs) and Catalase (CAT) ELISA Kit were purchased from Shanghai Xinyu Biotechnology Co., Ltd. (Shanghai, China); TRI Reagent Solution, RevertAid First Strand cDNA Synthesis Kit and SYBR™ Select Master Mix were purchased from ThermoFisher Scientific (ThermoFisher Scientific, Waltham, MA, USA); methane-sulfonate salt (MS-222) were purchased from Sigma (St. Louis, MO, USA); PBS was purchased from Sangon Biotech (Shanghai, China) Co., Ltd. (Shanghai, China). All the other reagents were analytically pure and made in China. The synthesis and sequencing of primers were operated by Guangzhou Tian Yi Hui Yuan Gene Technology Co., Ltd. (Guangzhou, China).

Ultrasound Breaker (SONICS, Newtown, CT, USA), Transmission Electron Microscope JEM 2100 (JOEL, Toyko, Japan), Laser Particle Size and Zeta Potential AnalyserZetasizer Nano S90 (MALVERN, Malvern, UK), Light Incubator (EYELA, Toyko, Japan), Spectra Max M2 (Molecular Devices Corp., Sunnyvale, CA, USA), Small Level Electrophoresis (Bio-Rad, Hercules, CA, USA), Applied Biosystems StepOnePlus^TM^ Real-Time System (ABI, Foster City, CA, USA).

The zebrafish (*Danio rerio*) used in the experiment were purchased from non-exposed adults (AB strain, aged 3 months). Adult fish were half male and half female (2.82 ± 0.31 cm in length, 0.48 ± 0.05 g in weight), maintained under constant temperature (25–26 °C) for two months. The water for breeding was tap water filtered through PP cotton / activated carbon and oxygenated for more than 48 h. The heating rod regulated the water temperature in the range of 26 ± 0.5 °C (pH value 7.0–7.3). The oxygen pump increased oxygen, and the dissolved oxygen in water was 8.23 + 0.08 mg/L. Adult fish were placed under natural light for a period of 14/10 h and fed in the morning and evening with a commercial flake food No. 0 (Charoen Pokphand Group, Bangkok, Thailand).

### 2.2. Physical and Chemical Representation of TiO_2_ NPs

The physical and chemical properties of TiO_2_ NPs were tested via Transmission Electron Microscope, TEM and Dynamic Light Scattering, DLS, which were carried out in the testing center of Hainan University, China.

### 2.3. Acute Toxicity Experiment of Zebrafish Embryos

The water used in embryo cultivation and embryo exposure experiment was oxygenated saturated standard embryo culture medium (E3 solution: 5 mmol/L NaCl, 0.17 mmol/L KCl, 0.33 mmol/L CaCl_2_ and 0.33 mmol/L MgSO_4_, containing no methylene blue, and the pH value adjusted to about 7.2 with NaHCO_3_ solution.). Normally-developed fertilized embryos were selected and cultivated in a 24-hole plate, with each hole infilled with 3ml blank control E3 solution, regular TiO_2_ contrast solution (100 mg/L), TiO_2_ NPs exposure solution of different concentrations (10, 50 and 100 mg/L) and 4 fertilized embryos, 10 replicates for each concentration. Group experiment was completed 2 h after the passing out of fertilized eggs. The culture plate was sealed with a lid, and placed in an incubator with a temperature of 28 ± 0.5 °C, and under a light/dark period of 14/10 h. During the exposure process, the growth of the fertilized eggs was monitored every 12 h, keeping records of the death and incubation of the eggs within 0–96 hpf (hour postfertilization, hpf) and getting rid of the whitening and condensation eggs to prevent contamination. The test solutions were changed every 24 h, and mixed by ultrasonic for 30 min before use. The embryo acute toxicity experiment was replicated quintically, with each concentration containing 200 embryos. The embryo hatching rate = (hatched embryos/the number of zebrafish embryos in total) × 100%.

### 2.4. Exposure Experiment of Adult Zebrafish

The concentration of TiO_2_ NPs was set at 10 mg/L, 50 mg/L and 100 mg/L. Usual water and normal TiO_2_ (100 mg/L) were set as control groups. The exposure concentrations used were determined by our former study and summary of references. The short-term toxicity of nano-TiO_2_ particles is generally low to aquatic species in the order of mg/L [10]. The highest concentration applied in the current study was based on 25% of EC_50_ (Based on the malformation rate, 24 h EC_50_ of TiO_2_ NPs was 500 mg/L for the zebrafish embryos in the prepared experiments). The lowest concentration was based on 10 times of an environmental investigation concentration, considering that the exposure time is only a few days. Each concentration was filled with 5 L of test solutions and 20 adult zebrafish (half male and half female) and placed under a light/dark period of 14/10 h. The exposure experiment of adult fish lasted for 7 days during which fresh test solutions were renewed every day to ensure the stability of water and test solutions. 3 replicates were set for each experiment group. After the exposure experiment, 6 fish from each concentration was taken out and anatomized for liver, intestine and gill on ice. The tissue was immediately rinsed in pre-cooled PBS buffer and then weighed and homogenized. Homogenate samples were centrifuged at 12,000× *g* for 30 min at 4 °C and stored at −80 °C for further testing. The sample supernatant could be used to examine the variations of protein markers related to the antioxidative damages of zebrafish. Animal welfare and experimental procedures were carried out in accordance with the Guide for the Care and Use of Laboratory Animals (China National Standardizing Committee GB 14925-2010 and Ministry of Science and Technology of China, 2006), and were approved by the animal ethics committee of Hainan Medical University.

### 2.5. ELISA Test for Three Oxidative Damages of Zebrafish

Three key protein markers for oxidative damages were selected as evidence of oxidative damages on different tissues of zebrafish. The three molecular biomarkers were SOD, CAT and GSTs. Microplate Reader was used to test the amount of SOD, CAT and GSTs in the tissue of liver, intestine, and gill of zebrafishrespectively. Activities of SOD, CAT and GSTs were measured using ELISA kits (Xinyu Biotechnology Co., Ltd, Shanghai, China) according to the manufacturer’s instructions.

### 2.6. Expression of Antioxidant Response-Related Genes of Three Different Zebrafish Tissues

The gene sequences were obtained from NCBI (GeneBank) using Primer 5.0 for the design of corresponding primer. qRT-PCRprimers were designed based on the antioxidant response-related enzyme genes of zebrafish while *β-actin* was selected as reference gene. Before our mRNA expression experiment, we assessed the amplification efficiencies of primers and transcriptional stability of three candidate genes (*rpl8*, *18s*, *β-actin*) commonly used as reference genes for BPAF from exposure to a single compound. The results of the analysis showed that the *β-actin* was the most stable gene for TiO_2_ single treatment and *β-actin* was selected as the reference gene for the mRNA expression assay in this study. The mRNA expression of each target gene was normalized to *β-actin*. *β-actin* transcript was used to standardize the results by eliminating variations in mRNA and cDNA quantity, as it did not vary upon chemical exposure (data not shown) and was used as internal control. Primers for zebrafish *β-actin* gene was also selected based on our previous research and other published literature [11,12]. The *β-actin* gene encodes a structural protein of cytoskeleton, shows high stability in zebrafish tissues and cells. The RNA quality was examined by measuring the 260/280 nm ratios (1.94–2.07) and 1% agarose-formaldehyde gel electrophoresis with GoldView™ (SBS Genetech Co., Ltd, Beijing, China) staining. Primer sets used in qRT-PCR are described in Table 1. Total RNA was isolated from three frozen tissue samples (100 mg) of zebrafish in different treatment groups using RNA isolatertotal RNA extraction Reagent (TRI Reagent Solution, ThermoFisher Scientific, Waltham, MA, USA). To minimize DNA contamination in RNA preparations, 1 μl DNase I (1U) was add to an RNA sample (1 μg). The prepared RNA can be used as a template for reverse transcriptase. Total RNA (1–2 μg) was reverse-transcribed to generate the first-strand cDNA using RevertAid First Strand cDNA Synthesis Kit (ThermoFisher Scientific, Waltham, MA, USA) according to the manufacturer’s instructions. qRT-PCR was performed in the Applied Biosystems StepOnePlus^TM^ Real-Time Syetem (ABI, Foster City, CA, USA) using SYBR™ Select Master Mix (ThermoFisher Scientific, Waltham, MA, USA) in a 25 μl reaction with final primer concentration of 200 nM. The primer sequences are detailed in Table 1. The thermal profile used 95 °C 1 min, 40 × (95 °C × 15 s, 58 °C × 20 s, 72 °C × 20 s). *β-actin* was used as reference gene. qRT-PCR would be administered on SOD, CAT and GSTs to observe the quantity of expression. The relative mRNA expression levels of antioxidant response-related genes were analyzed using the 2^−ΔΔCt^ method [13].

### 2.7. Statistical Analysis of Data

The data of the experiment are presented as Mean ± SD, using GraphPad Prism 6.0 for data and graphical analysis. The comparison of data was conducted using one-way ANOVA followed by T Test examining the significant differences between the exposure groups and the control groups of different concentrations. The differences were considered significant at three different levels (* *p* < 0.05; ** *p* < 0.01; *** *p* < 0.001) relative to the controls. qRT-PCR is conducted by 2^−ΔΔCt^ method. The relative expression level of mRNA of the three antioxidant response-related target genes of the samples could be compared with the expression level of reference gene *β-actin* in correspondent samples.

## 3. Results

### 3.1. Physical and Chemical Representation of TiO_2_ NPs

Figure 1 presents the TEM characterization of bulk TiO_2_ (A) and TiO_2_ NPs (B, C). The usual diameter for TiO_2_ was 150–200 nm and 25–40 nm for TiO_2_ NPs, in accordance with TiO_2_ NPs purchased (with an average diameter of 25 nm). The 24 h TiO_2_ NPs slurry, observed by TEM (Figure 1C), was identified with precipitating and concentrating of nanoparticles. Some nanoparticles existed in the form of coacervate in the solution, with an average diameter range of 100 nm.

DLS results are presented in Table 2. The test results reflect the effective diameter of TiO_2_ NPs in the solution. The diameters of TiO_2_ NPs in E3 solution were larger than those obtained by TEM. With the increase of concentration, the diameters of TiO_2_ NPs were also increasing. In 24 h, the diameters of TiO_2_ NPs in the solution were larger than the diameters of TiO_2_ NPs in the same concentration in 0 h. The test results indicate that TiO_2_ NPs increasingly concentrate and precipitate in the bottom with the passage of time.

### 3.2. Acute Toxicity Experiment of Zebrafish Embryos

Figure 2 shows the cumulative hatching rate of zebrafish embryos for 96 hpf (A) and survival rate of zebrafish embryos. The embryos began to hatch from 36 hpf and became fully hatched until 96 hpf. Most of the embryos were hatched and developed during 48–72 hpf. Compared with different exposure groups, the hatching rate exhibited no significant changes and the differences were statistically insignificant. The survival rate for each exposure group for 96 hpf was over 90% with no sign of deformity and delayed hatching. The test results show that acute TiO_2_ NPs exposure under the experimental concentration exerted no significant influence on the hatching and deformity rate of zebrafish embryos.

### 3.3. Results For Oxidative Damage Experiment Of Zebrafish

As is indicated in Figure 3, the activity of three enzymes related to antioxidant damage (SOD, CAT and GSTs) in gill tissue of adult zebrafish growing for 7 days in different concentrations of TiO_2_ NPs exposure solution was measured. Compared with the blank control group and the conventional TiO_2_ group, the activities of three enzymes were decreased, especially in the high concentration group. The activities of CAT, SOD and GSTs were 48.7%, 38.2% and 50.3% of the normal control group, respectively, with a significant decrease in the activity of SOD. However, compared with the blank control group, the activities of the three antioxidant enzymes didn’t change too much in the conventional TiO_2_ group. Compared with the blank group and the conventional TiO_2_ group, the activities of CAT and GSTs in gill tissue were significantly different (*p* < 0.05). Compared with the blank group and the conventional TiO_2_ group, the activity of SOD in gill tissue was significantly different (*p* < 0.05) in the high concentration exposure group of TiO_2_ NPs (100 mg/L).

Figure 4 shows the activity changes of three antioxidant enzymes in zebrafish liver exposed to different concentrations of TiO_2_ NPs. As an important organ for detoxification, liver is also the main site for regulating redox metabolism. There are a large number of SOD, CAT and GSTs in liver tissue, which can eliminate reactive oxygen. Compared with the blank group and the conventional TiO_2_ group, the activity of three antioxidant enzymes in liver tissue decreased at low concentration of TiO_2_ NPs, and with the increase of the concentration of TiO_2_ NPs in the exposed solution, the effects on the activity of three enzymes increased gradually. The activities of CAT, SOD and GSTs in the 100 mg/L TiO_2_ NPs exposure group were 32.3%, 26.22% and 40.2%, respectively, of those in the control group, each experiencing significant decrease. For the activity of three enzymes in liver tissue, the activities of CAT, SOD and GSTs in the three exposure concentration groups of TiO_2_ NPs were significantly lower (*p* < 0.05) than those in the conventional TiO_2_ exposure group and the blank group.

Figure 5 shows the activity of three antioxidant enzymes in zebrafish intestine after exposure to different concentrations of TiO_2_ NPs for 7 days. Compared with gill and liver tissues, the activities of antioxidant enzymes in intestine tissue were not significantly affected by TiO_2_ NPs, but were increased under high concentration of TiO_2_ NPs. Increased activity indicates that in the face of the toxicity of TiO_2_ NPs, intestine tissue can cope with the adverse reactions caused by oxidative stress by increasing the content of antioxidant enzymes to avoid oxidative damage to intestinal tissues and organs. In addition, the small intestine tissue may not be significantly affected due to the fact that after ingesting into the small intestine, TiO_2_ NPs are not absorbed by the small intestine, and therefore have little effect on the intestine cells.

### 3.4. Expression Analysis Of Oxidative Response Related-Genes in Three Tissues Of Zebrafish

Zebrafish were isolated from five exposure groups, and corresponding tissues (gill, liver and intestine) were separated to extract Total RNA. The extracted RNA samples were detected by agarose gel electrophoresis (Figure 6). Electrophoresis bands were clear, and 28 S, 18 S and 5 S were obvious. The total amount of Total RNA extracted from intestine tissue was less than that of gill and liver. The purity and concentration of Total RNA were further detected by ultraviolet spectrophotometer. The results are shown in Table 3. All the light absorption values of RNA (A260/A280) were around 2.0, which proved that the purity of RNA was good, and there was no protein and DNA contamination, meeting the needs of subsequent in vitro reverse transcription experiments. The corresponding cDNA was obtained by in vitro reverse transcription of the extracted Total RNA as template.

The quantitative real-time reverse transcription PCR was used to analyze the expression of SOD, CAT and GSTS genes in different zebrafish tissue samples. From Figure 7, we can see that the gene expression levels of three antioxidant enzymes in gill tissue of zebrafish increased after 7 days of exposure to conventional TiO_2_ and TiO_2_ NPs. Among them, SOD and GSTS genes were more obvious. The relative expression levels of CAT, SOD and GSTS genes in gill tissue were 7.13 ± 0.57, 2.34 ± 0.14 and 4.18 ± 0.64 respectively for high concentration exposure solution, which were 7.1, 2.3 and 4.2 times higher than those in blank control group. These results suggest that zebrafish can increase the expression of antioxidant enzymes by up-regulating the related antioxidant enzyme genes in order to resist adverse external stimuli after long-term exposure to high concentration of TiO_2_ NPs. In addition, the relative expression of CAT and GSTs genes was also up-regulated in the conventional exposure solution of TiO_2_, which were 2.84 ± 0.53 and 2.24 ± 0.16, respectively, and the gene expression was increased by about two times.

Figure 8 shows the expression of three oxidative damage related genes (CAT, SOD and GSTS) in liver tissue of zebrafish. All three genes were expressed in the exposed groups with different concentrations of TiO_2_ NPs, and up-regulated in the high concentration groups. The relative expression levels of CAT, SOD and GSTS genes in high concentration exposure group were 7.77 ± 0.85, 3.70 ± 0.37 and 3.27 ± 0.07, respectively. The relative expression levels of the three genes were 7.8, 3.7 and 3.3 times respectively in the blank control group. In the conventional TiO_2_ exposure group, CAT, SOD and GSTS genes were highly expressed at 3.30 ± 0.37 and 3.00 ± 0.15 respectively, about three times as much as those in the blank control group.

Figure 9 shows the expression of three oxidative damage related genes (CAT, SOD and GSTS) in intestine of zebrafish. The relative expression levels of CAT, SOD and GSTS genes in different concentrations of TiO_2_ NPs exposure solutions were all up-regulated. In the high concentration group, the relative expression level of CAT, SOD and GSTS were 4.75 ± 0.15, 1.66 ± 0.09 and 1.50 ± 0.22 respectively, which were 4.8, 1.7 and 1.5 times higher than those in the blank control group. In the conventional TiO_2_ exposure group, the expression of CAT and SOD genes in intestine tissue was up-regulated, and the relative expression levels were 2.54 ± 0.26 and 1.65 ± 0.05, which were 2.5 and 1.7 times higher than those in the blank control group, respectively.

## 4. Discussion

With the rapid development of nanotechnology, nanomaterials are widely used in many fields such as chemical industry, electronics, medicine, cosmetics and so on. As an important nanomaterial, nano TiO_2_ has been widely used as coatings, cosmetics, functional fibers, fine ceramics, catalysts, etc. [14]. In this study, the toxicity of TiO_2_ NPs on zebrafish embryos and adult fish is systematically studied. Before this study, research had already been carried out on the toxicity of TiO_2_ NPs to zebrafish embryos, but there are some differences between the experimental results and the toxic dose [15,16]. This study found that TiO_2_ NPs did not affect the hatching of zebrafish embryo in the concentration range of 100 mg/L. This is consistent with the findings of Vicario-Pares [17]. The disparity in the experimental results may be caused by the different crystal forms and particle sizes of TiO_2_ NPs, resulting in various toxicity effects. In addition, besides physical and chemical properties, the dispersion and precipitation rates of TiO_2_ NPs in solution are also important factors affecting their toxicity [18]. The short-term exposure of TiO_2_ NPs has no obvious toxicity to zebrafish embryos. It may be that the embryo eggshell prevents the TiO_2_ NPs from entering the embryos of zebrafish. There is no direct contact between TiO_2_ NPs and intraembryonic cells in the process of growth. The nutrients of embryo are mainly maintained by yolk sac. No external nutrients are needed, avoiding exchange with external substances and therefore exerting no obvious effect on embryo growth. Although there was no significant effect of TiO_2_ NPs on the growth of zebrafish embryo, it was reported that the acute toxicity of TiO_2_ NPs could be increased by external physical and chemical factors such as ultraviolet light [19]. In addition, the bioavailability and toxicity of some toxic substances, such as decabromobiphenyl ether (BDE-209), perfluorooctanesulfonate (PFOS), and pentachlorophenol (PCP), can be increased by using TiO_2_ NPs as carriers. When these substances interact with TiO_2_ NPs, the toxicity to zebrafish shall greatly increase [18,20,21,22].

In addition to acute embryonic toxicity, the long-term toxicity of TiO_2_ NPs was also studied in adult zebrafish. Researchers used adult zebrafish to study the enrichment and removal of anatase and gold-red crystalline TiO_2_ NPs in zebrafish under long-term exposure conditions. The results showed that after long-term exposure, there was a certain degree of enrichment of TiO_2_ NPs in zebrafish, but no sign of bioaccumulation. TiO_2_ NPs could be effectively discharged. Moreover, TiO_2_ NPs was mainly distributed in the digestive tract after ingestion through the mouth of zebrafish, and excreted through feces [23]. Similar phenomena were identified in the anatomical process of adult fish in the experiment. There was obvious accumulation of TiO_2_ NPs in the intestine tissue of zebrafish. Studies have shown that nanometal oxide particles can produce excessive reactive oxygen species (ROS) under light or some other conditions. Excessive ROS can destroy the antioxidant mechanism of the body itself by affecting cell communication and interfering with cell metabolism, thus producing toxic effects on organisms [24]. In addition, G. Federici et al. found that TiO_2_ NPs could cause lipid peroxidation damage in gill of rainbow trout (*Oncorhynchus mykiss*) [25]. Based on the previous findings, we also used oxidative damage as an important indicator of TiO_2_ NPs toxicity.

Oxidative stress is the excessive production of reactive oxygen species (ROS) and reactive nitrogen free radicals (RNS) in organisms when they are exposed to harmful stimuli. When the amount of these free radicals exceeds the scavenging capacity of the defense system, it will lead to imbalance of oxidation/antioxidant system and oxidative damage of organism tissue, such as lipid peroxidation, enzyme inactivation, DNA breakage and so on. SOD, CAT and GSTs are three important enzymes involved in oxidative stress in organisms. Their changes in activity and content reflect the balance of oxidation/antioxidant system in organisms [26]. SOD is a specific enzyme for scavenging superoxide anion (O_2−_), which can disproportionate with O_2−_, and decompose it into H_2_ and O_2_. SOD plays an important role in the balance of oxidation and antioxidation, and is one of the most important antioxidant enzymes. CAT is a marker enzyme of peroxisome, which can promote the decomposition of H_2_O_2_ into O_2_ and H_2_O. It accounts for about 40% of the total peroxisome enzymes. It is one of the key enzymes in biological defense system. It can effectively reduce the content of ROS and protect cells from the damage of H_2_O_2_. CAT has high content in liver and a fast decomposition rate. GSTs are important metabolic enzymes transformed in organisms, and are the main detoxification system for cell anti-oxidative damage and anti-cancer. Gill tissue is the organ that zebrafish directly contacts with the exposed solution of TiO_2_ NPs, while liver is the main place to regulate redox metabolism, and intestinal tract, which is an important organ that integrates secretion, immunity and barrier functions, is the main place for fish to absorb nutrients and to immunize. Therefore, we focused on the activities of enzymes related to oxidation/antioxidation and the expression of corresponding genes in these three tissues. ELISA testing results showed that the activities of three enzymes in gill and liver tissues were significantly lower than those in control group, and the differences of three oxidase activities were significant compared with those in blank control group and conventional TiO_2_ group. However, in intestine tissue, low and medium concentrations of TiO_2_ NPs did not cause activity change in the three enzymes related to oxidative damage. In the high dose group, there were slight changes in the activity of the three enzymes. The oxidative damage induced by excessive ROS is an important mechanism of nanomaterials poisoning. Excessive ROS can affect biological redox system (glutathione, thioredoxin, etc.), destroy the oxidative balance, change the activity of related enzymes, and lead to oxidative damage [27]. Gill tissue was directly exposed to TiO_2_ NPs solution, and TiO_2_ NPs produced a large amount of ROS under illumination [28]. Normally, SOD, CAT and GSTs in vivo can remove ROS and maintain the oxidative balance. However, excessive accumulation of ROS results in insufficient antioxidant enzymes in vivo, and ROS exceeds the body’s self-scavenging capacity, which will lead to the imbalance of the body’s oxidative and antioxidant systems [29,30]. The three oxidases can produce adaptive induction reaction due to mild exposure, and can also be inhibited by toxic reaction due to excessive ROS pollution [30]. Therefore, the gill tissue was greatly influenced by ROS and the activity of oxidase decreased. The liver is the main site for regulating redox metabolism. A large number of SOD, CAT and GSTs in the liver can remove excessive reactive oxygen species. TiO_2_ NPs can enter liver tissue through osmosis and blood circulation. In addition, TiO_2_ NPs can also promote ROS production by affecting cell metabolism and cell-to-cell interaction. Excessive ROS (•OH) may damage the body’s own antioxidant mechanism, thus causing toxicity to organisms [29]. Long-term exposure to TiO_2_ NPs leads to the accumulation of ROS, including O_2−_ and peroxide free radicals. Excessive production of ROS can damage the redox system of zebrafish, inhibit the activity of antioxidant enzymes, reduce their content and cause oxidative damage [31,32]. However, unlike ZnO NPs, TiO_2_ NPs had little effect on oxidative damage of zebrafish intestine tissue and did not cause damage to intestine tissue. Many studies show that treatment of cells with TiO_2_-NPs induces an increase in ROS production and oxidative products, such as lipid peroxidation, increased levels of GSH-Px, CAT, and SOD, as well as the depletion of cellular antioxidants [33,34,35]. Many studies have reported that TiO_2_-NPs induce toxic response by increasing the generation of ROS, which is associated closely with cell damage such as cell proliferation, inflammation, cell death and cell organelle disruption, indicating that ROS are also important signaling modulators [36,37]. There are studies that show lymphocytes decreased after subchronic exposure to high TiO_2_-NP concentrations suggesting immune functions, such as production of antibodies and recognition of antigens, might be affected [38]. The increase in GST activity, in turn, may be due to the increase in the biotransformation process of GSH and NP conjugation. A similar effect of TiO_2_-NP was found in tilapias, whose liver increased GST expression [39].

In order to further investigate the effects of TiO_2_ NPs on the oxidative damage of the three tissues, we examined the effects of TiO_2_ NPs on the expression of three antioxidant enzymes at the gene level. The gene expression of three antioxidant enzymes was up-regulated in different concentrations of exposure solution. By up-regulating gene expression, more enzymes were produced to resist the adverse effects of TiO_2_ NPs. This is also consistent with the basic defense mechanism of organisms. However, due to long-term exposure, an excessive amount of TiO_2_ NPs accumulates a large amount of ROS. Although gene expression increases, the corresponding antioxidant enzymes translated by mRNA of the three increased enzymes are still not enough to offset excessive ROS, resulting in oxidative damage in zebrafish. This further illustrates that TiO_2_ NPs can cause oxidative damage to the gill and liver of zebrafish under long-term exposure.

## 5. Conclusions

(1)Acute exposure to TiO_2_ NPs at experimental concentration has no significant effect on the hatchability and deformity rate of zebrafish embryos.(2)Under long-term exposure, TiO_2_ NPs can cause oxidative damage to the gill and liver tissues of zebrafish.(3)Under long-term exposure, the expression of CAT, SOD and GSTs antioxidant enzymes in zebrafish is up-regulated by TiO_2_ NPs to combat adverse reactions.

## Figures and Tables

**Figure 1 ijerph-16-00523-f001:**
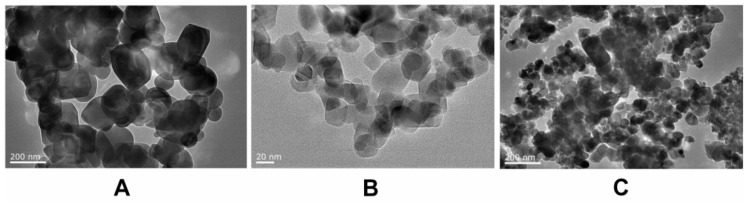
TEM Characterization of bulk TiO_2_ (**A**) and TiO_2_ NPs (**B**,**C**) TiO_2_ NPs exhibited good monodispersity and showed approximately normal distribution.

**Figure 2 ijerph-16-00523-f002:**
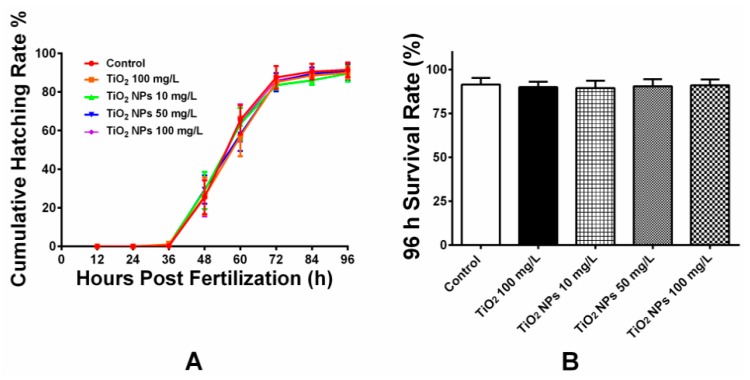
Hatching and survival rate of zebrafish embryos. (**A**) Hatching rate of zebrafish embryos induced by E3 solution, bulk TiO_2_ and different concentration TiO_2_ NPs for 0–96 hpf. The results showed no strong inhibition of hatching rate after embryos exposed to TiO_2_ nanoparticles; (**B**) Percentage of surviving embryos after 96 h of exposure. Results are presented as mean ± SD from five independent experiments (*n* = 200, 40 embryos in 5 replicates each).

**Figure 3 ijerph-16-00523-f003:**
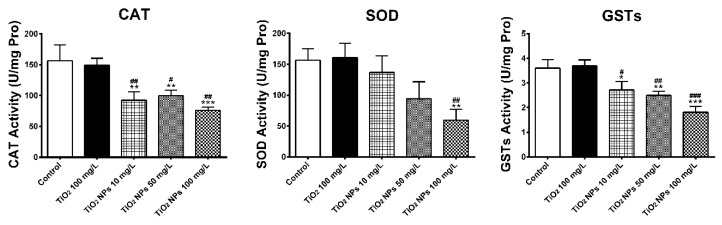
CAT, SOD and GSTs activities in zebrafish gill tissues. Data are expressed as means ± SD (*n* = 3). *, ** and *** indicate statistically significant differences from control values at *p* < 0.05, *p* < 0.01 and *p* < 0.001, respectively. #, ## and ### indicate statistically significant differences from TiO_2_ (100 mg/L) values at *p* < 0.05, *p* < 0.01 and *p* < 0.001, respectively.

**Figure 4 ijerph-16-00523-f004:**
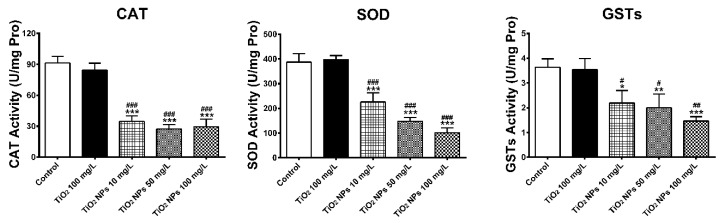
CAT, SOD and GSTs activities in zebrafish liver tissues. Data are expressed as means ± SD (*n* = 3). *, ** and *** indicate statistically significant differences from control values at *p* < 0.05, *p* < 0.01 and *p* < 0.001, respectively. #, ## and ### indicate statistically significant differences from TiO_2_ (100 mg/L) values at *p* < 0.05, *p* < 0.01 and *p* < 0.001, respectively.

**Figure 5 ijerph-16-00523-f005:**
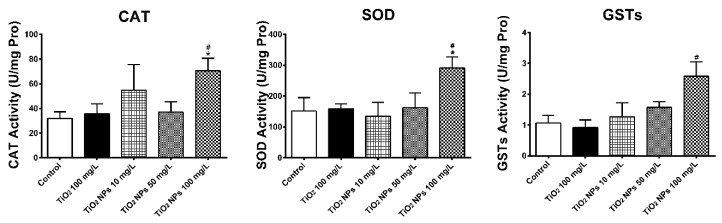
CAT, SOD and GSTs activities in zebrafish intestinetissue. Data are expressed as means ± SD (*n* = 3). * indicate statistically significant differences from control values at *p* < 0.05. # indicate statistically significant differences from TiO_2_ (100 mg/L) values at *p* < 0.05.

**Figure 6 ijerph-16-00523-f006:**
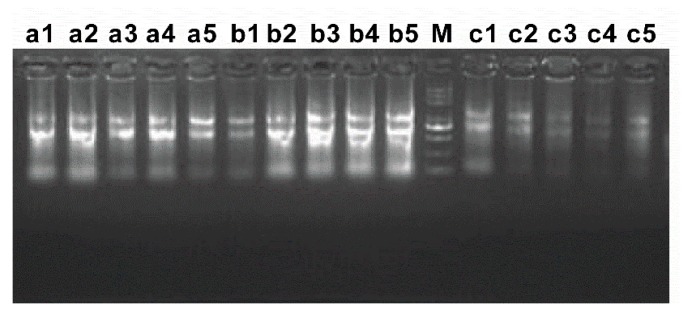
Agarose gel electrophoresis of RNA of zebrafish three tissues. a1–a5 gill tissues; b1-b5 liver tissues; c1–c5 intestine tissues; 1–5 represent different group.

**Figure 7 ijerph-16-00523-f007:**
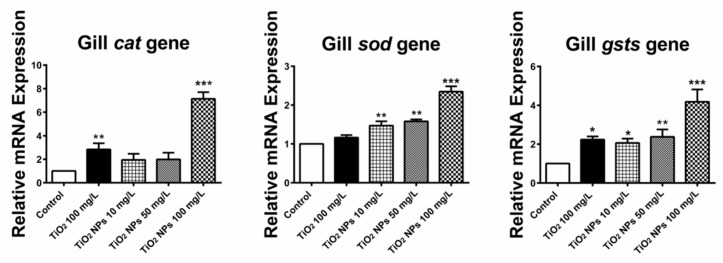
Change in the relative expression level of three antioxidase related genes (CAT, SOD and GSTS) in gill tissue. *, ** and *** indicate statistically significant differences from control values at *p* < 0.05, *p* < 0.01 and *p* < 0.001, respectively.

**Figure 8 ijerph-16-00523-f008:**
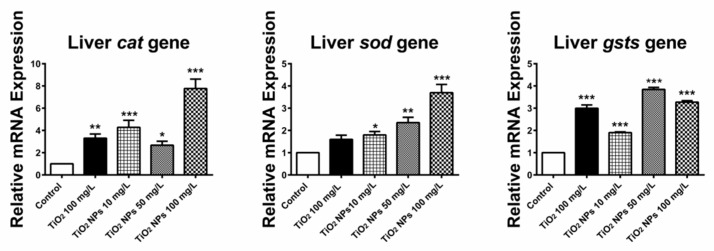
Change in the relative expression level of three antioxidant related genes (CAT, SOD and GSTS) in livertissue. *, ** and *** indicate statistically significant differences from control values at *p* < 0.05, *p* < 0.01 and *p* < 0.001, respectively.

**Figure 9 ijerph-16-00523-f009:**
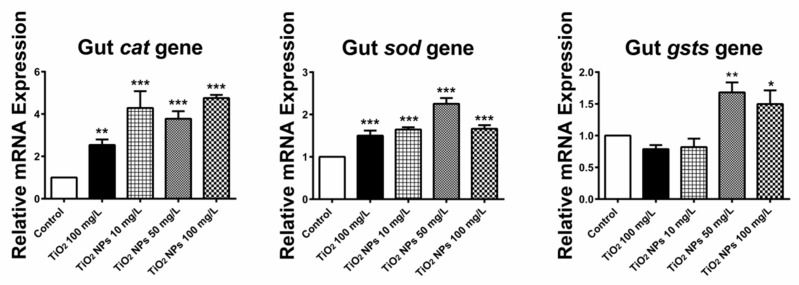
Change in the relative expression level of three antioxidant related genes (CAT, SOD and GSTS) in intestine tissue. *, ** and *** indicate statistically significant differences from control values at *p* < 0.05, *p* < 0.01 and *p* < 0.001, respectively.

**Table 1 ijerph-16-00523-t001:** Primer sequences used for amplification of antioxidant response-related genes with the qRT-PCR.

Gene	Accession Number	Forwand Primer (5′–3′)	Reverse Primer (5′–3′)	Amplicon (bp)
*Cu/Zn-sod*	NM_131294.1	CAAGAGGGTGAAAAGAAGCCA	GGTCACATTACCCAGGTCTCC	201
*cat*	NM_130912.2	AAGTCACTCACGACATCACGC	CGGTGTAGAACTTCACTGCGA	159
*gst*	NM_131734.3	CCCTCCTGTCTGGACTCTTTC	CAGTTTCTTGAAGTTCTCGCA	105
*β-actin*	NM_131031.2	CACAGTGCTGTCTGGAGGTAC	CATTTAAGGTGGCAACAGTTC	265

**Table 2 ijerph-16-00523-t002:** Hydrodynamic size and Zeta potential of TiO_2_ NPs in dispersion media.

Group	0 h		24 h	
	Diameter (nm)	Zeta Potential (mV)	Diameter (nm)	Zeta Potential (mV)
TiO_2_ (100 mg/L)	371.8 ± 27.4	−21.6 ± 1.1	466.8 ± 42.7	−27.2 ± 0.9
TiO_2_ NPs (10 mg/L)	101.4 ± 14.5	−22.8 ± 0.5	261.9 ± 22.1	−19.6 ± 0.7
TiO_2_ NPs (50 mg/L)	132.9 ± 8.1	−14.8 ± 0.5	327.6 ± 14.5	−21.5 ± 0.9
TiO_2_ NPs (100 mg/L)	140.4 ± 15.9	−17.7 ± 0.7	335 ± 29.9	−28.1 ± 1.1

**Table 3 ijerph-16-00523-t003:** The concentration of RNA from three different tissues.

Name of Sample	Concentration (ng/μl)	A260/A280
a1	1597.76	2.04
a2	1596.62	2.05
a3	773.00	1.99
a4	1423.89	2.03
a5	537.48	1.99
b1	1287.15	2.04
b2	1742.94	2.04
b3	2127.67	2.07
b4	1972.66	2.05
b5	2385.67	2.06
c1	486.77	1.96
c2	315.04	1.96
c3	270.62	1.96
c4	278.48	1.93
c5	247.95	1.94

a1–a5 gill tissues; b1–b5 liver tissues; c1–c5 intestine tissues.

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
