# Peer review of "Toxic Effects of TiO2 NPs on Zebrafish"

_ijerph, 2019, doi:10.3390/ijerph16040523_

Reviewer 1 Report

The authors study the effect of Titanium dioxide nanoparticles (TiO2 NPs) on zebrafish. TiO2 NPs are a widely used nanomaterial owing to their photocatalytic activity and absorption of UV light. This study investigated the toxic effects of rutile TiO2 NPs on zebrafish by examining effects in embryos and adults. Exposure to 100 mg/L TiO2 NPs didn’t affect the hatching rate of zebrafish embryos, and deformity in embryo morphology was not noted. However, when adults were examined, the effect of TiO2 NPs on oxidative damage in liver, intestine and gill tissue was more pronounced. ELISA and Q-PCR were used to detect three antioxidant enzymes: superoxide dismutase (SOD), catalase (CAT) and glutathione S transferase (GSTs) at both protein and gene levels which is a really nice aspect of this work. Long-term exposure to TiO2 NPs caused oxidative damage to zebrafish, and the decline of enzymatic activity. At the mRNA level TiO2 NPs caused high expression of cat, sod and gsts mRNAs in three organs as a counteractive response. The effects in liver and gill were more obvious given their roles in osmoregulation/detoxification.

This is a nice study by Tang and colleagues – I would recommend for publication once the following issues are addressed.

Quality of English is generally good but the article still needs judicious editing preferably by a native English speaker. There are some issues with Scientific English that need to be corrected throughout the article.

Why were the concentrations of the NPs set at 10, 50 and 100 mg/ml – there is not adequate justification for why these were the selected concentrations ?

What are the concentrations used in the cosmetic industry ? – are these concentrations lower or higher than what you would expect in cosmetics and sunblock – these details would provide greater context to the study.

Details on the Q-PCR need to be improved. With the aim of providing transparent research and promoting rigor and transparency - further details need to be provided for the Q-PCR assays - QPCR assays - so that the data is MIQE compliant. Were the designed assays exon spanning ? - How can the authors be sure the amplicons derive from RNA and not genomic DNA ? – no DNAse treatment of RNA is mentioned. The authors do not state if melting curves were performed or not – this data could be included in supplemental materials. Authors should note the The MIQE Guidelines: Minimum Information for Publication of Quantitative Real-Time PCR Experiments http://clinchem.aaccjnls.org/content/55/4/611, and make sure that the data is compliant.

Why was b-actin used as a housekeeping gene – were the enzyme mRNAS in the same abundance class ? Please provide justification.

When referring to mRNAS OR genes – the convention is to use italics – the authors should ensure consistency through the article.

The ethical concern I have relates to whether an IACUC - i.e. an Institutional Animal Care and Use Committee (IACUC) or equivalent ensured Humane Care and Use of Laboratory Animals ?. Once they hatch (larvae typically hatch from their chorion or egg shell between 3-4 days post fertilization (dpf)) they are considered as vertebrates. 

Author Response

Dear reviewer,

Thank you for your comments concerning our manuscript ijerph- 426825. Those comments are all valuable and very helpful for revising and improving our paper, as well as the important guiding significance to our researches. We have studied comments carefully and have made correction which we hope meet with approval. The manuscript ijerph-426825 has been carefully revised and some new data were added in the revised text and the major revisions were marked in red.

In this study, this information would be helpful in providing a further theoretical basis for ecological risk assessment. The new manuscript was improved the description of the method that needs to be more precise (the major revisions were marked in red).

Details Reference Annex

Reviewer 2 Report

The study was well designed and the manuscript should consider the following points:

• In the results, they refer to the immunohistochemical study, so they should add their micrographs. As in the methodology refer to the immunohistochemical techniques used the respective antibodies.

• In the discussion of the results should attempt to explain the levels of hepatic toxicity to TiO2 NPs, referring to membrane / cytoplasmic receptors that may be involved in this inflammatory process and consequent hepatotoxicity.

Author Response

Dear reviewer,

We are very grateful to your comments for our study. The manuscript ijerph-426825 has been carefully revised and some new data were added in the revised text and the major revisions were marked in red.

We appreciate the detailed and useful comments and suggestions from you and referees. The point-by-point answers to the comments and suggestions were listed as below:

Details Reference Annex

Round  2

Reviewer 1 Report

I commend the authors for their detailed responses to my critiques. In my opinion, the manuscript is now suitable for publication if the authors incorporate their very nice responses to my queries into the manuscript - or alternatively provide as supplemental information where appropriate.

Q1: Why were the concentrations of the NPs set at 10, 50 and 100 mg/L – there is not adequate justification for why these were the selected concentrations?

Q2: What are the concentrations used in the cosmetic industry? –Are these concentrations lower or higher than what you would expect in cosmetics and sunblock -these details would provide greater context to the study.

Q3: Details on the Q-PCR need to be improved. 

Q4: Why was β-actin used as a housekeeping gene – were the enzyme mRNAS in the same abundance class? Please provide justification.

Author Response

Dear reviewer,

    We are very grateful to your comments for our study. The manuscript ijerph-426825-R1 has been carefully revised and some new data were added in the revised text and the major revisions were marked in Green. 
